# Recipient Pericardial Apolipoprotein Levels Might Be an Indicator of Worse Outcomes after Orthotopic Heart Transplantation

**DOI:** 10.3390/ijms25031752

**Published:** 2024-02-01

**Authors:** Andrea Székely, Éva Pállinger, Evelin Töreki, Mandula Ifju, Bálint András Barta, Balázs Szécsi, Eszter Losoncz, Zsófia Dohy, Imre János Barabás, Annamária Kosztin, Edit I. Buzas, Tamás Radovits, Béla Merkely

**Affiliations:** 1Department of Anesthesiology and Intensive Therapy, Semmelweis University, 1085 Budapest, Hungary; 2Heart and Vascular Center, Semmelweis University, 1085 Budapest, Hungary; 3Department of Genetics, Cell- and Immunobiology, Semmelweis University, 1085 Budapest, Hungary; eva.pallinger@gmail.com (É.P.);; 4Faculty of Medicine, Semmelweis University, 1085 Budapest, Hungary; 5Doctoral School of Theoretical and Translational Medicine, Semmelweis University, 1085 Budapest, Hungary; szecsibalazs14@gmail.com (B.S.);; 6HCEMM-SU Extracellular Vesicle Research Group, Semmelweis University, 1085 Budapest, Hungary; 7HUN-REN-SU Translational Extracellular Vesicle Research Group, Semmelweis University, 1085 Budapest, Hungary

**Keywords:** organ recipient, heart transplantation, primary graft dysfunction, interleukin, apolipoprotein

## Abstract

Background: End-stage heart failure (ESHF) leads to hypoperfusion and edema formation throughout the body and is accompanied by neurohormonal and immunological alterations. Orthotopic heart transplantation (HTX) has been used as a beneficial option for ESHF. Due to the shortage of donor hearts, the ideal matching and timing of donors and recipients has become more important. Purpose: In this study, our aim was to explore the relationship between the clinical outcomes of HTX and the cytokine and apolipoprotein profiles of the recipient pericardial fluid obtained at heart transplantation after opening the pericardial sac. Materials and methods: The clinical data and the interleukin, adipokine, and lipoprotein levels in the pericardial fluid of twenty HTX recipients were investigated. Outcome variables included primer graft dysfunction (PGD), the need for post-transplantation mechanical cardiac support (MCS), International Society for Heart and Lung Transplantation grade ≥2R rejection, and mortality. Recipient risk scores were also investigated. Results: Leptin levels were significantly lower in patients with PGD than in those without PGD (median: 6.36 (IQR: 5.55–6.62) versus 7.54 (IQR = 6.71–10.44); *p* = 0.029). Higher ApoCII levels (median: 14.91 (IQR: 11.55–21.30) versus 10.31 (IQR = 10.02–13.07); *p* = 0.042) and ApoCIII levels (median: 60.32 (IQR: 43.00–81.66) versus 22.84 (IQR = 15.84–33.39); *p* = 0.005) were found in patients (*n* = 5) who died in the first 5 years after HTX. In patients who exhibited rejection (*n* = 4) in the first month after transplantation, the levels of adiponectin (median: 74.48 (IQR: 35.51–131.70) versus 29.96 (IQR: 19.86–42.28); *p* = 0.039), ApoCII (median: 20.11 (IQR: 13.06–23.54) versus 10.32 (IQR: 10.02–12.84); *p* = 0.007), and ApoCIII (median: 70.97 (IQR: 34.72–82.22) versus 26.33 (IQR: 17.18–40.17); *p* = 0.029) were higher than in the nonrejection group. Moreover, the pericardial thyroxine (T4) levels (median: 3.96 (IQR: 3.49–4.46) versus 4.69 (IQR: 4.23–5.77); *p* = 0.022) were lower in patients with rejection than in patients who did not develop rejection. Conclusion: Our results indicate that apolipoproteins can facilitate the monitoring of rejection and could be a useful tool in the forecasting of early and late complications.

## 1. Introduction

Orthotopic heart transplantation (HTX) has been used as a beneficial option for end-stage heart failure (ESHF) [1,2,3]. Improvement in the selection criteria, more accurate determination of urgency, and optimization of preoperative conditions have helped to decrease adverse outcomes [4]. Nevertheless, HTX still has a considerable complication rate, including primer graft dysfunction (PGD), early graft loss, rejection, and mortality. A recent comparison of available risk scores raised concern about their discriminative ability for one-year mortality [5].

In contrast, several biomarkers have been used in risk estimation. These diagnostic tests are part of the routine pre-transplantation evaluation of factors and include natriuretic peptides, the glomerular filtration rate, bilirubin levels, etc. [4]. Interleukins (ILs) and apolipoproteins (Apo) have also been suggested to play important roles in the development and progression of ESHF [6,7]. These new biomarkers seem to have promising prognostic roles in the development of diabetes and atherosclerosis. Only sparse data exist regarding the application of these new markers in transplantation medicine [8]. Based on the known donor shortage, the optimization of patient selection, use of aged donors in borderline indications, and adding these new biomarkers to recipient selection might help to improve the risk prediction of the available risk scores.

We hypothesized that the extent of immunological injury can be quantified and that we can obtain a more accurate picture of heart function if aliquots are sampled from pericardial fluid. The aim of this study was to compare the levels of different interleukins and apoproteins in the pericardial fluid with the possible adverse outcomes of recipients after HTX. The primary outcome was primary graft dysfunction. Mortality and rejection were also investigated. Additionally, the inotropic score (IS), vasoactive-inotropic score (VIS), United Network for Organ Sharing (UNOS) recipient risk score, and Index for Mortality Prediction After Cardiac Transplantation (IMPACT) score were correlated with pericardial hormone levels.

## 2. Results

### 2.1. Recipient Characteristics

The median age of the recipients was 58.5 years, and all the recipients were male (*n* = 20). There was gender mismatch in 20% (*n* = 4) of the HTXs, and the medians of the recipients’ and donors’ body mass index (BMI) values were 25.6 and 25.05 kg/m^2^, respectively. The etiology of heart failure was ischemic in 75% (*n* = 15) and idiopathic in 25% (*n* = 5) of the transplants. Further descriptive characteristics of the recipients are summarized in Table 1.

### 2.2. Recipient Risk Scores and Laboratory Parameters

The median scores of the patients were 1.00 (IQR: 1.00–2.25) and 5.00 (IQR: 2.00–9.00) for the UNOS recipient score and IMPACT score, respectively. The UNOS-R score showed association with the adipsin (*p* = 0.020) and adiponectin (*p* = 0.008) levels based on the ANOVA test. The IMPACT scores were not related to the measured panels. The ApoAII levels were higher in patients who had bilirubin levels above 1 mg/dL compared to those who had values < 1 mg/dL (*p* = 0.035). ApoAI showed a correlation with preoperative creatinine, aspartate aminotransferase *(*AST), and alanine transaminase (ALT) levels (r = −0.489, r = 0.615, r = 0.604; and *p* = 0.040, *p* = 0.004, *p* = 0.005, respectively). Preoperative AST levels were also correlated with the ApoCII (r = 0.462, *p* = 0.040) and ApoCIII (r = 0.463, *p* = 0.040) levels. Preoperative bilirubin levels were correlated with the IL-21 (r = 0.470, *p* = 0.002), IL-6 (r = 0.651, *p* = 0.001), IL-9 (r = 0.669, *p* < 0.001), and adiponectin (r = 0.690, *p* = 0.001) levels. Further correlations of the laboratory parameters are summarized in Appendix A.

### 2.3. Complications and Adverse Reactions

After HTX, five patients needed postoperative MCS, and four patients had PGD. Four patients had ISHLT Grade 2R rejection, and vasoplegia was diagnosed in four cases. The rejection rate from all the performed endomyocardial biopsies through the first year after HTX was 19.7%. The detailed postoperative complications and laboratory parameters are shown in Table 2.

In the pericardial fluid of the recipients, leptin levels were significantly lower in patients with PGD than in those without it (*p* = 0.029). Leptin levels were also significantly lower in the MCS group (*p* = 0.042). The detailed comparisons are shown in Table 3, Appendix A, Figure 1.

In the recipient pericardial fluids, ApoD levels were significantly higher in patients who had postoperative vasoplegia than in those who had no vasoplegia (median: 28.69 (15.85–37.66) vs. 12.34 (8.92–20.34) µg/mL, *p* = 0,022 in the vasoplegic and nonvasoplegic patients, respectively) (Appendix A, Figure 2).

Five-year mortality was 25% (5 patients). The mean survival time was 5.85 years (95% C.I.: 4.74–6.97 years). Two patients died in the first year, one patient in the second year, and two patients between the third and fifth post-transplantation years. Higher ApoCII (*p*= 0.042) and ApoCIII (*p* = 0.005) levels were detected in patients (*n* = 5) who died in the first 5 years after HTX (*n* = 5) (Table 4 and Appendix A; Figure 3).

In the first month after HTX, five (antibody-mediated rejection) AMR 1R and four AMR 2R were found. In patients who had ISHLT grade ≥ 2R rejection (*n* = 4) in the first month after transplantation, adiponectin (*p* = 0.039), ApoCII (*p* = 0.007), and ApoCIII (*p* = 0.029) levels were higher than in the nonrejection group, while pericardial T4 levels were lower in patients with rejection than in those who did not develop rejection. The data are shown in Table 5 and Appendix A, Figure 4.

### 2.4. Vasoactive-Inotropic Score and Inotropic Score

Significant correlations were found between the maximum vasoactive-inotropic score, inotropic score, and several immunological parameters. The detailed results are shown in Figure 5 and Figure 6. As can be seen, the correlations were significant between the inotropic scores but not the vasopressor scores.

### 2.5. K-means Clustering Based on The Molecular Composition of the Pericardial Fluid

K-means clustering (Figure 6 and Appendix A) identified three subgroups of immunological parameters (cluster A: Adipsin, Leptin, T4, IL-21, IL-6, IL-9, and oxLDL; cluster B: T3, ApoD, ApoAII, ApoCIII, ApoCII, ApoB100, ApoAI, ApoM, and adiponectin; cluster C: IL-4, IL-17A, IL-2, IL-13, IL-17F, IFN, TNF, IL-5, IL-10, and IL-22) and of patients (clusters 1–3). The clustered proteins (clusters A-C) show a similar distribution of values among the patients, indicating parallel shifts in expression in patients of the same cluster (clusters 1–3), which were automatically detected by the unsupervised algorithm. Patients of cluster 1 showed overall elevated levels of the interleukin-rich cluster C, while cluster 3 was characterized by higher degrees of pericardial apolipoprotein content (cluster B). Furthermore, patient clusters 1 and 3 demonstrated numerically greater incidences of MCS, PGD, and mortality (2/3, 2/3, 1/3 and 3/6, 2/6, 4/6, respectively) compared to the patients of cluster 2 (0/11, 0/11, 0/11).

## 3. Materials and Methods

### 3.1. Study Design, Setting, and Participants

Between January 2013 and April 2017, 206 transplantations were performed and 118 recipient pericardial fluid specimens were available (57.3% of the transplantations). Well-characterized pseudonymized human pericardial fluid and cell-depleted pericardial fluid samples of 20 randomly chosen recipients from the period between February 2013 and December 2017 were obtained from the Transplantation Biobank of the Heart and Vascular Center at Semmelweis University, Budapest, Hungary. Pericardial fluid samples contaminated with blood were part of the exclusion criterion in the study. The procedure for sample procurement was reviewed and approved by the institutional and national ethics committee (ethical permission numbers: ETT TUKEB 7891/2012/EKU [119/PI/12.] and ETT TUKEB IV/10161-1/2020). Clinical patient data were obtained from the database of the Transplantation Biobank. Our study was conducted in accordance with the Eurotransplant standards for organ sharing and with the Hungarian National Blood Transfusion Service. The last check on the follow-up data was made on 22 January 2023 [9].

### 3.2. Local Protocols

Donor and recipient variables were retrieved from the National and Eurotransplant-based donor data report form and from electronic medical records from our institutional databank. A diagram of the study procedure is shown in Figure 7.

### 3.3. Major Study Parameters

Due to the relatively small sample size of our study, the UNOS score was calculated for donors, recipients, and overall individuals. The donor-specific UNOS (UNOS D) score includes donor age (1 point if aged between 50 and 55 years or 2 points if aged above 55 years), total ischemic time (above 4 h: 2 points), sex mismatch (1 point), and donor diabetes mellitus (1 point). According to these criteria, the UNOS D score was calculated, and three risk groups were formed: low- (score: 0), intermediate- (score: 1 or 2), or high- (score: ≥3) UNOS D risk groups. The recipient-specific UNOS score considered the following parameters: age (above 65 years: 1 point), body mass index (30–35 kg/m^2^: 1 point; >35: 2 points), mean pulmonary artery pressure (above 30 mmHg: 1 point), total bilirubin (1.5–1.9 mg/dL: 1 point; >1.9 mg/dL: 2 points), creatinine (1.5–2.0 mg/dL: 1 point; >2 mg/dL: 2 points), previous transplant, previous cancer, and pre-transplant mechanical ventilation (each 2 points) or mechanical circulatory support (noncontinuous-flow: 2 points) [10]. The total score is the sum of donor- and recipient-specific scores, and this score was used in the multivariable logistic regression analyses for adjustment.

#### IMPACT Score

The IMPACT score included the following 12 recipient characteristics: age (above 60 years: 1 point), serum bilirubin (0–0.99 mg/dL: 0 points; 1–1.99 mg/dL: 1 point; 2–3.99 mg/dL: 3 points; ≥4 mg/dL: 4 points), creatinine clearance (≥50 mL/min: 0 points; 30–49 mL/min: 2 points; <30 mL/min: 5 points), dialysis between listing and transplant (4 points), female sex (3 points), heart failure etiology (idiopathic: 0 points; ischemic: 2 points; congenital: 5 points; other: 1 point), recent infection (3 points), intra-aortic balloon pump (3 points), mechanical ventilation pre-transplant (5 points), race (Caucasian: 0 points; African American: 3 points; Hispanic: 0 points; other: 0 points), temporary circulatory support (7 points), and ventricular assistance device (older-generation pulsatile: 3 points; newer-generation continuous: 5 points; Heartmate II: 0 points) [11].

### 3.4. Sample Collection and Preparation

After sternal splitting and opening of the pericardium, ACD (trisodium citrate with citric acid and dextrose) vacutainer tubes (BD Vacutainer^®^ System, BD Biosciences, San Jose, CA, USA) were used for the collection of pericardial fluid samples. After pelleting cells (300 g, 10 min, 4 °C), the supernatant was centrifuged at 2000 g for 10 min at 4 °C to remove large particles. The cell-free supernatants were collected, divided into three 500 µL aliquots, and stored in liquid nitrogen until use.

### 3.5. Flow Cytometric Multiplexed Bead-Based Immunoassays

LEGENDplex™ bead-based immunoassays (BioLegend, San Diego, CA, USA) were used for the quantification of pericardial fluid cytokines, adipokines, and apolipoproteins according to the manufacturer’s instructions.

### 3.6. Enzyme-Linked Immunosorbent Assays (ELISA)

Pericardial fluid oxidized low-density lipoprotein (oxLDL), triiodothyronine (T3), and thyroxine (T4) concentrations were determined via ELISA (human OxLDL ELISA^®^ Kit of Cloude Clone; T3 and T4 Human ELISA Kits Abcam, Cambridge, UK) according to the manufacturer’s instructions. Oxidative damage of pericardial fluid proteins was assessed by determining the free SH (sulfhydryl) content expressed in cysteine equivalents via the DTNB (5,5′-dithio-bis-(2-nitrobenzoic acid = Ellman reagent) method (Thermo Scientific Pierce Ellman’s Reagent, Thermo Scientific, Norristown, PA, USA).

### 3.7. Outcomes

Our primary outcome was primary graft dysfunction (PGD) in recipients. PGD was defined according to the consensus criteria of the International Society of Heart and Lung Transplantation (ISHLT) [12]. The decision regarding mechanical circulatory support (MCS) implantation was made by a team of experts (including a cardiologist, a cardiac surgeon, and a cardiac anesthesiologist) according to international guidelines [13]. Acute rejection was defined as an event that necessitated increased immunosuppression with an ISHLT grade ≥ 2R endomyocardial biopsy result or with noncellular reactions with hemodynamic compromise. Patients underwent routine surveillance for allograft function via endomyocardial biopsy and echocardiography at 1, 2, 3, and 4 weeks, as well as at 3, 6, 9, and 12 months after transplantation [14]. Post-transplantation vasoplegia was defined according to the ISHLT criteria [15]. Mortality was also assessed, and survival was checked on 30 April 2023.

The maximum vasoactive-inotropic score (VIS) and the length of administration were also analyzed. VIS was calculated using vasopressor (norepinephrine, epinephrine, vasopressin) and inotropic (dobutamine, dopamine, levosimendan, milrinone) medication doses. For the calculation of IS, the medication doses of dopamine, dobutamine, and epinephrine were used [16].

### 3.8. Statistical Analysis

Data are expressed as the median and interquartile range (first–third (IQR)). Differences between the groups were assessed using the nonparametric Mann‒Whitney U test for continuous variables. The association between outcomes and inflammatory variables was tested via Spearman’s correlation. Comparisons between ILs, apolipoproteins, and recipient scores were conducted by performing 1-way analysis of variance on the ranks test. Data were analyzed by using IBM-SPSS 22.0 software (International Business Machines Corporation, Armonk, New York, NY, USA). K-means clustering was performed in R 4.3.2 (R Foundation for Statistical Computing, Vienna, Austria) using the ComplexHeatmap package and Z-score normalization [17], utilizing the algorithm of Hartigan and Wong [18] with the number of expected clusters defined as 3. Elements of the resulting clusters were further clustered hierarchically. All of the statistical tests were two-sided, and a *p* value < 0.05 was considered to indicate statistical significance.

## 4. Discussion

We found that recipients who subsequently had PGD or required ventricular assistance device support had lower pericardial leptin levels than recipients who did not. Higher pericardial ApoD levels were associated with vasoplegia, and higher adiponectin, ApoB100, ApoCII, and ApoCIII levels and lower T4 levels were associated with rejection. Furthermore, higher pericardial ApoCII and ApoCIII levels showed a significant correlation with mortality.

The pathophysiology of ESHF is complex, characterized by volume overload, different extents of end organ dysfunction neurohormonal changes, and inflammation [19]. Several risk scores have been developed to determine the urgency of HTX or for usage of MCS before transplantation to bridge the unstable state or severe hypoperfusion to candidacy [11,20,21]. Systemic inflammation and cytokine release can further worsen hemodynamic instability and thus can serve as a diagnostic tool in risk estimation. With the help of K-means clustering, we created three groups according to high inflammatory reaction, high apolipoprotein levels, and those without these features. In these two clusters we found increased occurrence of early postoperative adverse events, such as PGD, need for MCS, and vasoplegia. This algorithm helped us to create groups, which were not labeled in the database or the relationship would have remained undiscovered without using this statistical method. Application of this method in HTX can be severalfold; for instance, in the matching, outcome, and follow-up of the patients [22,23]. We aim to use this method extensively in the future in a large population. Therefore, this study presents preliminary, exploratory results regarding the possible link between apoliporoteins and recipient outcomes after HTX.

K-means clustering indicated that simultaneous elevated levels of IL-4, IL-17A, IL-2, IL-13, IL-17F, IFN, TNF, IL-5, IL-10, and IL-22 may be associated with worse clinical outcomes by recognizing their similar distribution among patients in an unsupervised manner. Myocardial damage activates the innate and the adaptive immune system through the release of pro- and anti-inflammatory cytokines, coordinating the inflammatory response of the heart [24,25]. Although cluster C (Figure 7) includes cytokines widely regarded as either pro- or anti-inflammatory mediators, their simultaneous increase is part of the development of inflammation. Pro- and anti-inflammatory processes do not unfold in a vacuum. Instead of the predominance of one or the other, their dynamic balance regulates or dysregulates the inflammatory state.

The current challenges of HTX are the increased age of donors and recipients, the large number of patients who have MCS or are operated on in high-urgency status, and increased allosensitization [26,27]. The occurrence of early and late complications has also increased. One component of Apo B-100 produced by hepatocytes is found on the surface of every atherogenic particle [28,29]. The circulating level is an effective measure of cardiovascular risk and coronary artery disease [29,30]. A postmortem study revealed that the pericardial fluid of people with severe atherosclerosis contained higher amounts of ApoB than those without atherosclerosis [31].

Adiponectin is mainly produced by adipose tissue, and low levels of adiponectin are associated with ischemic heart diseases and peripheral arterial diseases. However, a high adiponectin level does not always mean a better outcome, because patients with severe cardiovascular disease in addition to liver or kidney disease often have elevated levels due to insufficient clearance as a secondary consequence of cardiovascular disease [32]. Elevated adiponectin levels might contribute to HTX in patients with nonischemic cardiomyopathy.[33]. Leptin is a nonglycosylated protein produced mainly by adipose tissue that plays an important role in normal cardiovascular function [32]. In young, healthy men, there is an inverse correlation between the thickness of the carotid wall and the circulating leptin concentration, which indicates the vascular protective effect of leptin [34]. In patients with nonischemic cardiomyopathy, lower leptin levels are associated with an increased likelihood of HTX [33]. In our study population, low leptin levels were associated with PGD and the need for MCS support. A post-transplantation need for high inotropic and vasoactive support has been found to be associated with 5-year mortality and hemodialysis [35]. The strong association between the inotropic score and pericardial ApoAI, ACII, ApoCIII, and ApoM levels might be explained by the local inflammatory environment, which can lead to myocardial dysfunction, or in more severe cases to PGD and MCS support.

ApoD has recently been the subject of research on neurodegenerative diseases and cardiovascular function. Circulating apoD levels have been observed to be increased in heart failure patients, and apoD plays an important role in inflammation [36,37]. The occurrence of vasoplegia in the post-transplantation period is 4.2–28.7% [13]. The origin is multifactorial (long cardiopulmonary bypass time, pre-transplantation device therapy, reoperation, bleeding, etc.), but acute and chronic inflammatory reactions contribute to the process. ApoD levels were significantly elevated in cases of vasoplegia, which might be a marker of inflammatory vasodilatation but must be tested in a larger population.

ApoCIII is a small 8.8 kDa apolipoprotein produced in the liver and found on the surface of very-low-density lipoprotein (VLDL), low-density lipoprotein (LDL), and high-density lipoprotein (HDL). ApoCIII has both indirect atherogenic effects (it causes hypertriglyceridemia) and direct atherogenic effects (it stimulates monocyte adhesion to endothelial cells, promotes the production of inflammatory mediators in these cells, and increases LDL retention in the arterial wall) [38,39]. ApoCIII had crucial regulatory role in the lipid metabolism of the liver, particularly in the triglyceride-rich lipoprotein transport pathways [40]. High ApoCIII levels have been measured in diabetic patients, in chronic kidney disease, and in patients with non-alcoholic fatty liver disease [41]. The measurement of ApoCIII levels might be used preoperatively to estimate the composite severity scores of metabolic diseases in parallel with kidney and liver functions [42]. We also found correlations among ApoCIII, GFR, and elevated transaminase levels. These coexisting diseases will not be cured with HTX. Moreover, chronic kidney disease and diabetes might become more severe. Therefore, ApoCIII screening might also be used in the post-transplantation period. ApoCII is mainly produced in the liver and interacts with the systemic inflammatory environment [43]. While there are fewer publications relating to ApoCII compared with studies on ApoCIII, our results indicate that the two apolipoproteins have similar kinetics in postoperative adverse events.

Despite the successful HTX therapy used in ESHF, allograft rejection still remains an issue with morbidity and mortality. Endomyocardial biopsy and staging of rejection have been standardized in recent decades, but the procedure itself can have procedural complications and it also has diagnostic limitations. The occurrence rate is 12% in the first transplantation year. Rejection has become more important in the current era, as older patients, previous MCS, previous pregnancies, and transfusion frequently lead to allosensitization. New molecular diagnostic approaches have been developed in the last decade, such as microarray technology, cell-free DNA, microRNA, and gene expression profiling [44]. In our study population, among the interleukins and apolipoproteins, adiponectin and ApoCII and ApoCIII levels were elevated in patients with AMR > 2R. After HTX, the therapy should be adjusted based on the prediction of rejection episodes (maintained alloreactivity), prognosis of allograft damage progression, and personal drug response [45]. The ideal immunosuppression would be a noninvasive strategy that can reliably discriminate between the presence and absence of rejection and overimmunosuppression [46].

### Limitations of the Study

The main limitation of this study is that the pericardial fluid samples of the donors were not available. There are no data regarding the serum levels. Moreover, the small population size and the heterogeneity of the recipients resulted in lower statistical power.

## 5. Conclusions

As the number of patients on waiting lists exceeds the potential of HTX, new prognostic scores or more sensitive biomarkers are needed to determine the optimal candidates for HTX. Our results indicate that early post-transplant complications were associated with marked differences in apolipoprotein levels, particularly with the elevation of ApoCII and ApoCIII levels in cases of mortality and rejection. We also found that with the cluster analysis we could further focus on associations between postoperative complications, therapeutic interventions (e.g., high vasoactive support), markers of end-organ function (e.g., low GFR, high bilirubin levels), and lipoproteins. Based on our results, we suggest preoperative and postoperative screening of ApoCIII and ApoCII levels, as they might be considered contraindications for HTX or as integral components of the UNOS classification.

## Figures and Tables

**Figure 1 ijms-25-01752-f001:**
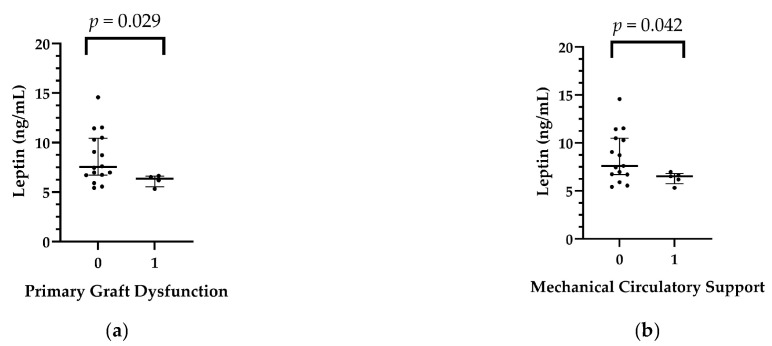
(**a**) Relationship between recipients’ pericardial leptin levels and primary graft dysfunction after heart transplant. (**b**) Relationship between recipients’ pericardial leptin levels and postoperative mechanical circulatory support after heart transplant.

**Figure 2 ijms-25-01752-f002:**
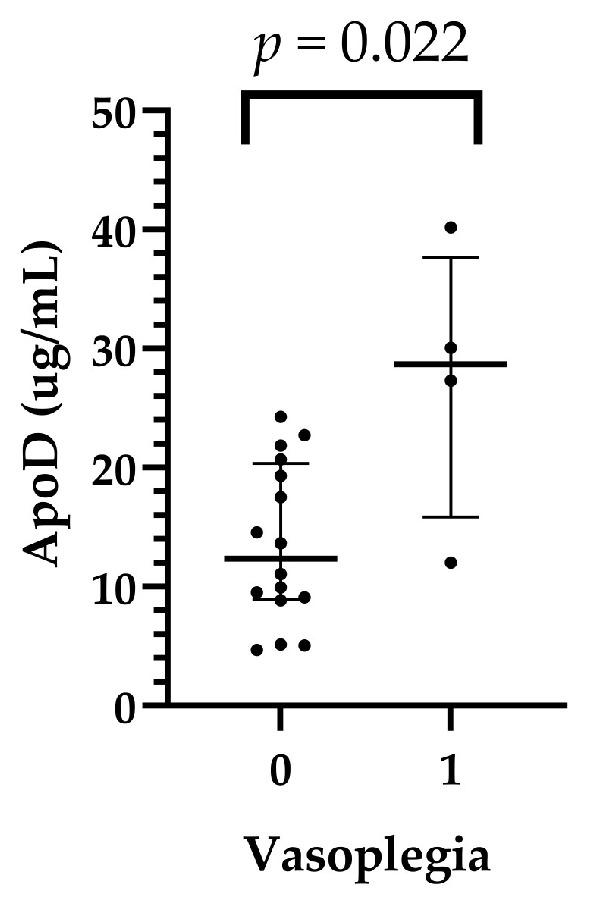
Relationship between recipients’ pericardial ApoD levels and vasoplegia after heart transplant.

**Figure 3 ijms-25-01752-f003:**
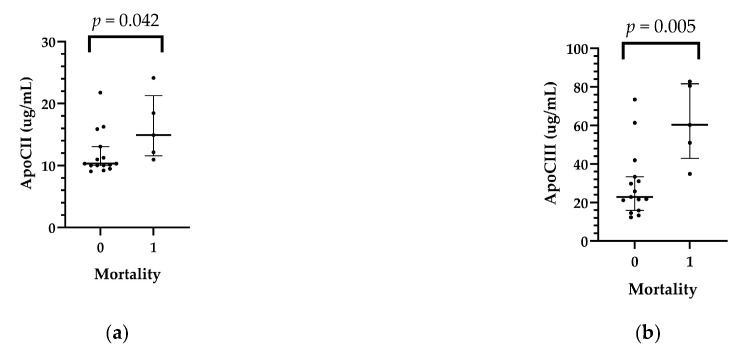
Relationship between recipients’ pericardial ApoCII (**a**) and ApoCIII (**b**) levels and 5-year mortality.

**Figure 4 ijms-25-01752-f004:**
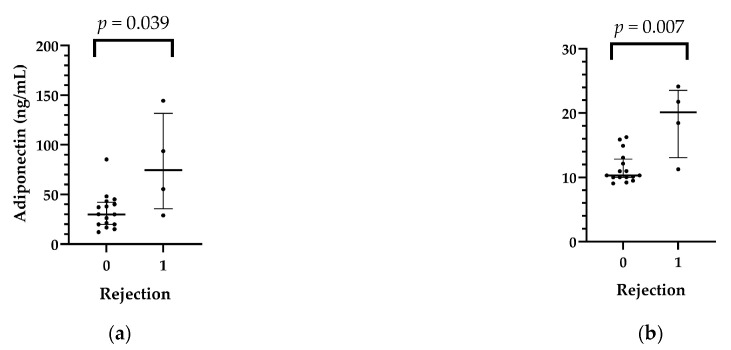
Relationship between recipients’ pericardial adiponectin (**a**), ApoCII (**b**), ApoCIII (**c**), and T4 (**d**) levels and rejection after heart transplant.

**Figure 5 ijms-25-01752-f005:**
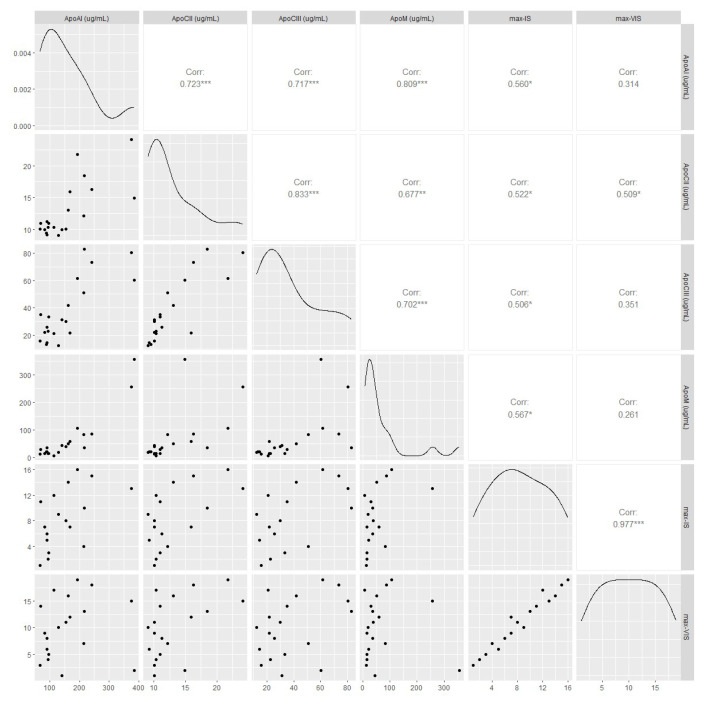
Correlations between vasoactive-inotropic score, inotropic score, and apolipoproteins. * *p* < 0.05, ** *p* < 0.01, *** *p* < 0.001.

**Figure 6 ijms-25-01752-f006:**
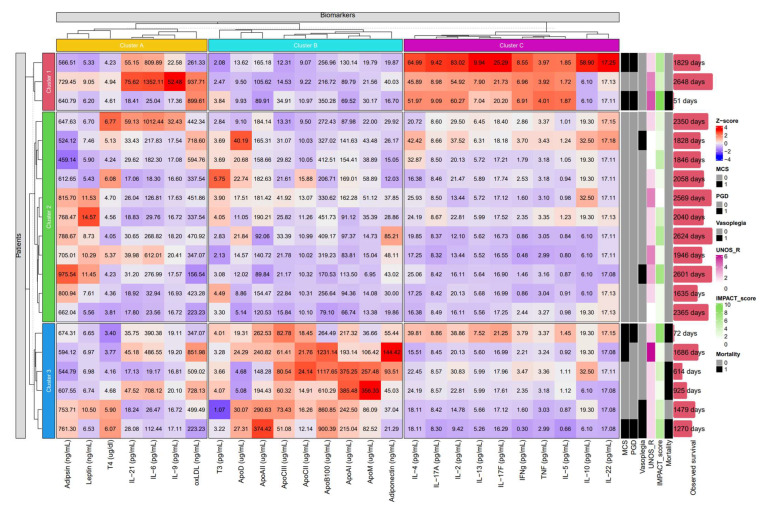
Heatmap with K-means clustering based on immunological parameters of the pericardial fluid of HTX recipients. Three main clusters of patients (clusters 1-3) and of proteins (clusters A–C) were identified. With respect to clinical outcomes, two patients in cluster 1, a cluster characterized by increased expression of cytokines of cluster C, underwent MCS due to PGD, which proved fatal in one patient. Cluster 3, showing increased expression of numerous apolipoproteins (cluster B), included two cases of PGD and three cases of MCS. Patients of cluster 2 with lower expression of apolipoproteins and cytokines were spared from these major complications. Apo: apolipoprotein; IL: interleukin; IMPACT: Index of Mortality Prediction After Cardiac Transplantation; MCS: mechanical circulatory support; PGD: primer graft dysfunction; UNOS: United Network for Organ Sharing. In case of MCS, PGD, vasoplegia, and mortality, 0 (grey) signifies absence and 1 (black) indicates occurrence of the respective complication. Apo: apolipoprotein; Corr: correlation; IS: inotropic score; VIS: vasoactive-inotropic score.

**Figure 7 ijms-25-01752-f007:**
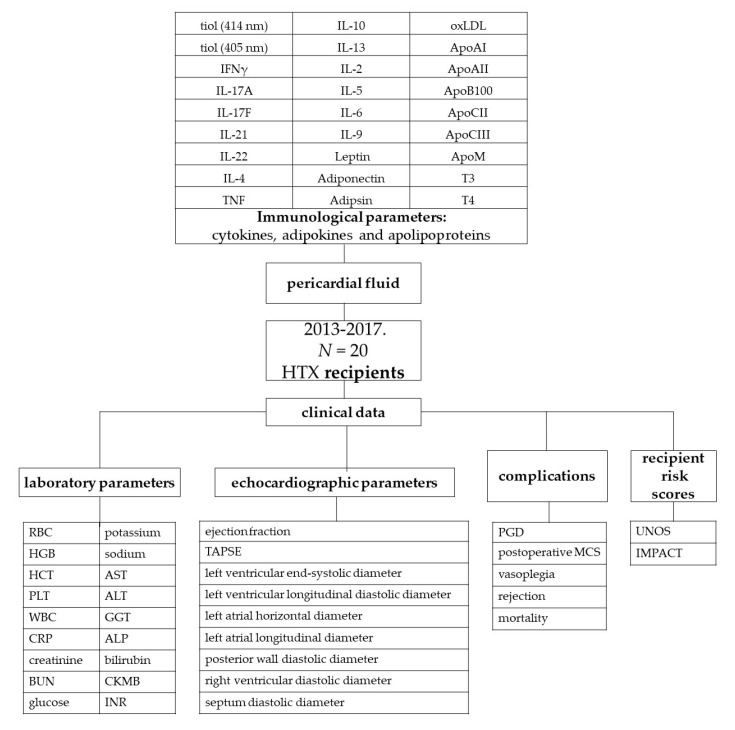
Diagram of the study procedure (ALP: alkaline phosphatase; ALT: alanine transaminase; Apo: apolipoprotein; AST: aspartate aminotransferase; BUN: blood urea nitrogen; CKMB: creatinine kinase-MB isoform; CRP: C-reactive protein; GGT: gamma-glutamyl transferase; HCT: hematocrit; HGB: hemoglobin; HTX: heart transplantation; IFN γ: interferon-γ; IL: interleukin; IMPACT: Index for Mortality Prediction After Cardiac Transplantation; INR: international normalized ratio; MCS: mechanical circulatory support; oxLDL: oxidized low-density lipoprotein; PGD: primary graft dysfunction; PLT: platelet count; RBC: red blood cell; T3: triiodothyronine; T4: thyroxine; TAPSE: tricuspid annular plane systolic excursion; TNF: tumor necrosis factor; UNOS: United Network for Organ Sharing; WBC: white blood cells).

**Table 1 ijms-25-01752-t001:** Descriptive characteristics of the recipients and heart transplantations.

Factor	*n* Median	% (IQR 25–75)
UNOS recipient score	1.00	(1.00–2.25)
0 1 2 3 4 5	3 11 1 3 1 1	15.00 55.00 5.00 15.00 5.00 5.00
IMPACT score	5.00	(2.00–9.00)
Demographic parameters		
Age Weight Height BMI Sex: Male	58.5 80 178 25.6 20	(53.00–60.00) (75.00–84.50) (173.00–182.00) (24.70–26.60) 100.00
Sex mismatch	4	20.00
NYHA		
III IV	6 14	30.00 70.00
Diagnosis		
Idiopathic heart disease Ischemic heart disease	5 15	25.00 75.00
Echocardiography parameters		
LVLDD LVLSD Posterior wall diastolic diameter Ascending aorta diameter Septum diastolic diameter Left atrial longitudinal diameter Left atrial horizontal diameter Right atrial longitudinal diameter Right atrial horizontal diameter Aortic root systolic diameter AoVmax	71.00 60.00 40.00 34.00 9.00 47.00 60.00 44.00 54.00 22.00 0.95	(61.75–78.25) (54.75–67.50) (37.00–44.00) (31.25–35.75) (7.00–9.75) (42.50–54.50) (51.50–63.50) (41.00–49.00) (43.00–63.00) (19.50–25.00) (0.70–1.10)
Left ventricular ejection fraction	21.1	(15.0–27.25)
Preoperative laboratory values		
Sodium (mmol/L)	138.00	(134.25–139.00)
Potassium (mmol/L) Creatinine (µmol/L)	4.20 102.00	(4.10–4.60) (96.25–151.00)
INR	1.79	(1.26–2.21)
AST (UI/L)	27.00	(21.50–31.25)
ALT (UI/L)	26.50	(16.50–33.00)
GGT (UI/L)	74.00	(47.25–159.25)
LDH (UI/L) ALP (UI/L) Total bilirubin (µmol/L) Total protein (g/L) Albumin (g/L) Cholesterol (mmol/L) Triglycerides (mmol/L)	349.00 92.50 11.45 72.10 45.55 4.10 1.08	(291.00–382.00) (68.75–107.25) (6.97–20.95) (68.32–75.65) (42.55–47.25) (3.65–5.40) (1.00–1.68)
Hemodynamic parameters		
Systolic pulmonary artery pressure	55.5	(45.0–62.25)
Diastolic pulmonary artery pressure	23.3	(17.25–30.25)
Mean pulmonary artery pressure	35.5	(30.0–42.5)
Pulmonary artery wedge pressure	23.1	(20.25–28.5)
Pulmonary vascular resistance (Wood units)	3.19	(2.42–4.13)
Cardiac output (L/min)	3.81	(3.0–4.6)

ALP: alkaline phosphatase; ALT: alanine transaminase; AST: aspartate aminotransferase; BMI: body mass index; GGT: gamma-glutamyl transpeptidase; IMPACT: Index for Mortality Prediction After Cardiac Transplantation; INR: international normalized ratio; IQR: interquartile range; LDH: lactate dehydrogenase; LVEDD: left ventricular end-diastolic diameter; LVESD: left ventricular end-systolic diameter NYHA: New York Heart Association; UNOS: United Network for Organ Sharing.

**Table 2 ijms-25-01752-t002:** Descriptive characteristics of complications and adverse reactions.

Factor	*n* Median	% (IQR 25–75)
Mortality		
5th year	5	25
Postoperative MCS	5	25
Perioperative complications		
Vasoplegia Primary graft dysfunction Rejection Reoperation Retransplantation	4 4 4 5 1	20 20 20 25 5
Transfusion		
RBC (units) Platelets (units)	6.50 3.50	(3.75–19.75) (3.00–12.75)

IQR: interquartile range; MCS: mechanical circulatory support; RBC: red blood cell.

**Table 3 ijms-25-01752-t003:** Relationship between recipients’ pericardial leptin levels and postoperative complications.

	No PGF		PGF		No MCS		MCS	
**Leptin (ng/mL)**	median	IQR (25–75)	median	IQR (25–75)	median	IQR (25–75)	median	IQR (25–75)
	7.54	(6.71–10.44)	6.36	(5.55–6.62)	7.61	(6.70–10.50)	6.53	(5.77–6.81)
*p* value	0.029				0.042			

IQR: interquartile range; MCS: mechanical circulatory support; PGD: primary graft dysfunction.

**Table 4 ijms-25-01752-t004:** Relationship between recipients’ pericardial ApoCII and ApoCIII levels and 5-year mortality.

	Did Not Die Within 5 Years		Died Within 5 Years		
	median	IQR (25–75)	median	IQR (25–75)	*p* value
ApoCII (ug/mL)	10.31	(10.02–13.07)	14.91	(11.55–21.30)	0.042
ApoCIII (ug/mL)	22.84	(15.84–33.39)	60.32	(43.00–81.66)	0.005

Apo: apolipoprotein, IQR: interquartile range.

**Table 5 ijms-25-01752-t005:** Relationship between recipients’ pericardial adiponectin, ApoCII, ApoCIII, and T4 levels and rejection.

	No Rejection		Rejection		
	median	IQR (25–75)	median	IQR (25–75)	*p* value
Adiponectin (ng/mL)	29.96	(19.86–42.28)	74.48	(35.51–131.70)	0.039
ApoCII (ug/mL)	10.32	(10.02–12.84)	20.11	(13.06–23.54)	0.007
ApoCIII (ug/mL)	26.33	(17.18–40.17)	70.97	(34.72–82.22)	0.029
T4 (ug/mL)	4.69	(4.23–5.77)	3.96	(3.50–4.46)	0.022

Apo: apolipoprotein; IQR: interquartile range; T4: thyroxine.

## Data Availability

The data presented in this study are available on request from the corresponding author. The data are not publicly available due to privacy reasons.

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
