# Peer review of "Recipient Pericardial Apolipoprotein Levels Might Be an Indicator of Worse Outcomes after Orthotopic Heart Transplantation"

_ijms, 2024, doi:10.3390/ijms25031752_

Round 1

Reviewer 1 Report (Previous Reviewer 2)

Comments and Suggestions for Authors

Dear authors,
Thank you for providing the revised version of your manuscript.

I think the paper improved after the corrections and is now suitable for publication.

Comments on the Quality of English Language

Minor editing of English language required

Author Response

Dear Reviewer,

Thank you for your all your comment, please see attached the point-by-point letter.

Many thanks

Reviewer 2 Report (Previous Reviewer 1)

Comments and Suggestions for Authors Abstract should be separated into the following parts: Background, Purpose and Materials and Methods, Results and Conclusion. In the "materials and methods" part it should be added that patients underwent heart transplantation, not just transplantation. I suggest adding a percentage of rejection from all performed EMBs. Heart transplantation should be shortened to HTx after the 1st mention and then there was no need to use the full definition in this article. I recommend authors to rewrite the following sentence: "Patients with nonischemic cardiomyopathy plus higher adiponectin levels are more likely to re-

ceive a transplant than those with low adiponectin levels [33]​".​ Adiponectin levels will not affect the doctor's decision who should be transplanted, the main criteria is the UNOS level in those who've been selected in the HTx WL.

​Please, add in the conclusion whether the level of adiponectin could be a contraindication for heart transplantation or a part of UNOS classification. Comments on the Quality of English Language

I suggest ask native English speaker for checking this article and switch some words into the UK English spelling.

Author Response

Dear Reviewer,

Thank you for your all your comment, please see attached the point-by-point letter.

Many thanks

This manuscript is a resubmission of an earlier submission. The following is a list of the peer review reports and author responses from that submission.

Round 1

Reviewer 1 Report

Comments and Suggestions for Authors While the topic of this article is important, the organization order in this manuscript did not make any sense and was not traditional. Why is the "materials and methods" section added after the "results"? Statistical analysis should be checked. You don't "median" for mentioning 1 patient from 20 with a particular score etc. It did not make sense to add the following information in the table that none of the patients had hypothyroidism but 1 was managed with L-thyroxine and then 2 patients with hyperthyroidism. None of them were managed for the low level of TSH? I suggest adding LVEF and PASP to the TTE results section. Please, make it clear: 10 patients out of 20 died in 5 years? Or others did not reach 5 years? Only 20 patients were transplanted in 5 years or 5 years ago there were 20 transplanted and then you followed-up this group? What did you mean under the definition "rejection"? Cellular or humoral or it was combined into 1 definition? Was it rejection with transplant dysfunction or just AMR2 with no DSA? Conclusion should be based on authors' results but not cited from other publications. Comments on the Quality of English Language

I suggest checking this article by a native English speaker.

Author Response

Dear Reviewer,

Please see attached our report to your comments.

Many Thanks

Reviewer 2 Report

Comments and Suggestions for Authors

In this paper, the authors aim was to explore the relationship between the clinical outcome of HTX and the cytokine and apolipoprotein profiles of the recipient pericardial fluid obtained at transplantation after opening the pericardial sac. They found that different levels of leptin, ApoCII, ApoCIII, adiponectin are pericardial tiroxin were correlated in different ways to post-transplantation outcomes in 20 patients.

Unfortunately I think that data are reported in a pretty confusing way that needs to be clarified before evaluating this paper for publication. Moreover I do not really understand what would be the clinical implication of these results and how these findings could improve clinical management.

In details, my concerns are following reported:

1.       I was not able to understand if the pericardial fluid you analyzed was the pericardial fluid of donors or of recipients. In the method section, you fist say that “Well-characterized pseudonymized human pericardial fluid and cell-depleted pericardial fluid samples of 20 randomly chosen recipients were obtained” and I thought that you were talking about recipients’ pericardial fluid. Then you state that “Between January 2013 and April 2017, 206 orthotopic transplantations were performed, and 118 donor pericardial fluid specimens were available (57.3% of the transplantations).” And moreover in the limitations you repeat that “The main limitation of this study is that the pericardial fluid samples of the recipients were not available”. I think this issue must be cleared because if you analyzed donors’ pericardial fluid, the title is misleading and all the paper must be analyzed differently.

2.       If you have data about the pericardial fluid characteristics of all heart transplants’ recipients in the Transplantation Biobank of the Heart and Vascular Center, why did you only choose 20 patients? And how did you choose them? In the paper you say “randomly”, but please specify how randomization was performed.

3.       Moreover, these 20 patients are not representative of the general population, for example they are all male. Why?

4.       In table 1 there is the line “bridge to transplant”. What do you mean? Do you mean patients on mechanical circulatory support as bridge to transplant? It is not clear. I suggest to remove the variable as there are 0 patients.

5.       In paragraph 2.2 of the results you report the correlations between laboratory parameters. Please specify if these correlations are positive or negative, otherwise I do not understand the point of reporting these data at all.

6.       Postoperative MCS rate and primary graft dysfunction rate are pretty high. Are these numbers representative of your general population?

7.       You have 5 reoperations out of 20 patients in your population and 1 retransplant. Please specify the reasons for reoperation

8.       In paragraph 2.4 you report that “Significant correlations were found between maximum vasoactive-inotropic score, inotropic score and several immunological parameters”, but the detailed results are available only in the supplementary materials. I think they should be included in the text or at least reported in a table, as this is part of you study results

9.       In the discussions you report that “Our results also indicate the strong relationship between the UNOS score and markers of kidney and liver dysfunction.” I do not understand where you demonstrated such correlation

10.   I think that the discussion section must be reviewed. In your results you report a correlation between apolipoproteins’ levels and outcomes after heart transplantation but you didn’t even try to give an explanation of this correlation.

11.   Moreover, you discuss about the opportunity to use these markers to “provide patient stratification and better immunosuppression treatment selection”. Please explain how.

Comments on the Quality of English Language

Minor editing of English language required

Author Response

(The authors gave the same response as above.)

Round 2

Reviewer 1 Report

Comments and Suggestions for Authors

Revised article looks a lot better.

However, there are few issues that should be revised: it didn't make sense to add 0 female.

I recommend again that the "materials and methods" chapter should be before the "results" chapter.

Author Response

Dear Rewiever,

Many thanks for all your comments again.

Hopefully we managed to revise the manuscipt finally.

Reviewer 2 Report

Comments and Suggestions for Authors

Dear authors,

thank you for providing the revised version of your manuscript.

I think the paper improved after the corrections and is now suitable for publication. Please provide the figure legends of the new figures that you added in the text.

Author Response

(The authors gave the same response as above.)
